# Changes in the Physicochemical Parameters of Yoghurts with Added Whey Protein in Relation to the Starter Bacteria Strains and Storage Time

**DOI:** 10.3390/ani10081350

**Published:** 2020-08-04

**Authors:** Aneta Brodziak, Jolanta Król, Joanna Barłowska, Anna Teter, Mariusz Florek

**Affiliations:** Institute of Quality Assessment and Processing of Animal Products, University of Life Sciences in Lublin, Akademicka 13, 20-950 Lublin, Poland; aneta.brodziak@up.lublin.pl (A.B.); joanna.barlowska@up.lublin.pl (J.B.); anna.wolanciuk@up.lublin.pl (A.T.); mariusz.florek@up.lublin.pl (M.F.)

**Keywords:** yoghurt, WPC, storage, water activity, texture, colour index, syneresis

## Abstract

**Simple Summary:**

Addition of whey proteins to yoghurt may be a good solution that could be routinely applied in the dairy industry to offer consumers a new functional product, with health-promoting properties. Conducting a comprehensive analysis of yoghurts made using two different starter cultures, with and without the addition of various levels of whey proteins, we found that the quality of yoghurts with WPC, including sensory quality, can be satisfactory for even 28 days of storage. The use of whey protein concentrate determined both the physicochemical (i.e., lactic acid content, proximate chemical composition, water holding capacity, water activity, firmness, consistency, cohesive strength and colour parameters) and sensory quality of the yoghurts. The additive had a significant effect on potential acidity, inhibiting the increase in the amount of lactic acid in yoghurts during storage, and also reduced syneresis. We suggest that using WPC on a larger scale will create new opportunities on the food market.

**Abstract:**

The stability of the physicochemical characteristics of yoghurts during refrigerated storage is important for industry and the consumer. In this study we produced plain yoghurts with the addition of health-promoting whey protein concentrate (WPC), using two different starter cultures based on *Streptococcus thermophilus* and *Lactobacillus delbrueckii* subsp. *bulgaricus*. Physicochemical changes (acidity, nutritional value, water activity, water-holding capacity, texture, and colour, including whitening and yellowing indices) as well as sensory changes occurring during 28-day refrigerated storage were determined. Starter cultures were found to significantly (*p* ≤ 0.05 and *p* ≤ 0.01) influence the water-holding capacity, firmness, consistency, cohesive strength and colour parameters of the curd. Use of whey protein concentrate affected both the physicochemical and sensory quality of the yoghurts. The additive had a significant effect on potential acidity, inhibiting the increase in lactic acid in the yoghurts during storage, and also reduced syneresis. However, it decreased the lightness of curd and negatively affected its sensory characteristics, primarily flavour. Moreover, nearly all parameters changed significantly with the passage of storage time (in most cases negatively). The exceptions were total protein and fat content. The changes, however, were not severe and remained at a level acceptable to tasters. Addition of 1% or 2% whey protein to yoghurt may be a good solution that can be routinely applied in the dairy industry to offer consumers a new functional product. A comprehensive assessment of the physicochemical and sensory changes occurring during refrigerated storage of yoghurts manufactured with the addition of WPC and using different cultures is crucial for modelling such a product.

## 1. Introduction

Dairy products are an important component of the human diet. Yoghurt is currently one of the most commonly consumed fermented milk beverages. Consumers value yoghurt for its flavour, as well as its nutritional value and health-promoting effects. Moreover, yoghurt has features of functional food, for which demand is continually growing [1,2]. Yoghurt is milk that is acidified and coagulated using bacteria of the species *Lactobacillus delbrueckii* ssp. *bulgaricus* and *Streptococcus thermophilus* bacteria [3]. It should contain at least 10^7^ live bacteria per g. The International Dairy Federation (IDF) has also approved the use of other bacterial cultures, such as *Bifidobacterium* and *Lactobacillus acidophilus*. The unique matrix of this category of food improves its quality and increases the amount of absorbed nutrients. Their synergistic effect supports the functioning of the body, which is particularly important in disease states, primarily gastrointestinal disorders [4,5].

With the development of knowledge in the field of food and human nutrition, milk and fermented milk products are increasingly shown to exert a beneficial effect on the body [6,7]. Although the beneficial properties of yoghurt have long been known, continuous efforts are currently being made to improve its nutritional value and its sensory and functional features, in part through the use of various added ingredients, such as fruits, cereals, seeds, fibre from fruits or vegetables, or milk proteins. The primary direction of research involves improving the consistency of yoghurts and reducing syneresis. For this purpose, various polysaccharides and milk proteins in the form of skimmed milk powder, whey powder, or whey protein concentrates (WPC) are added [8,9,10,11]. The addition of whey protein to yoghurt can significantly improve its rheological properties, extend its shelf life, and reduce production costs [9,11,12]. Whey proteins (consisting of individual proteins, i.e., α-lactalbumin, β-lactoglobulin, serum albumin, immunoglobulins, lactoferrin, lactoperoxidase and lysozyme) exert a well-documented positive effect on human health, including supporting the functioning of the cardiovascular, digestive and nervous systems, and demonstrating anticancer, antioxidative, antimicrobial (antibacterial and antiviral) and hypocholesterolemic effects. These proteins, as a uniquely rich and balanced source of amino acids (i.a., cysteine and tryptophan), possess a significant meaning for human nutrition. Occurrence of whey proteins exhibiting antimicrobial properties, i.e., immunoglobulins, lactoferrin, lactoperoxidase and lysozyme, in the diet is one of the factors determining the correct immune response. Moreover, these proteins, once partially digested, serve as a source of bioactive peptides with further physiological activities [12,13,14,15]. According to Glibowski and Rybak [12], the use of protein preparations in the dairy industry has made it possible to produce a non-fat version of yoghurt without sacrificing its desired organoleptic properties. Comparison of traditional yoghurt with yoghurt with added whey protein has shown that the addition of WPC improves the texture and homogeneity of the curd. Whey proteins used in yoghurt production result in a more homogeneous microstructure, characterized by a smaller pore diameter, which limits the migration of fluid from within the yoghurt to its surface. Importantly, the use of whey protein in the production of milk-based beverages helps to reduce the adverse phenomenon of syneresis. The addition of these substances increases water-holding capacity during storage; the liquid is not separated from the yoghurt, which thus retains valuable nutrients [16,17]. According to Lesme et al. [18], modifying the protein content also reduces the release of most flavour compounds, which can impact the overall aroma balance and change aroma perception. For industry and the consumer, however, what is important is the stability of these characteristics during refrigerated storage of yoghurts.

These are important issues due to the growing interest in yoghurt. According to the Institute of Agricultural and Food Economics [19], yoghurt production in Poland in the years 2016–2019 amounted to approximately 400,000 tonnes. For comparison, in Europe in the analogue period it was estimated at 8.2 mln tonnes for yoghurts and other products called acidified milk. The average annual consumption of yoghurt in individual Polish households has remained constant since 2014, at 6 kg per person (for comparison, it is over 20 kg in Europe). Purchases of yoghurt accounted for 12% of expenditures on dairy products. However, production of whey amounted 55 mln tonnes in EU-28 in 2018 [20]. It is, therefore, large scale and with great potential sector of the dairying. The main industrial processing of whey is drying (i.e., whey powder production), which is 70% of the annual production of whey. Small tonnage of WPC and to a lesser extent WPI (whey protein isolates) are produced every year, and the remaining permeate can be used to produce WPH (whey protein hydrolysates), protein fractions, lactic acid, bioethanol or lactose [21]. It is estimated that about 20% of the whey is converted into WPC.

The aim of the research was therefore to produce yoghurts with the addition of health-promoting whey proteins, using two different starter cultures, and to determine the physicochemical and sensory changes taking place during refrigerated storage.

## 2. Materials and Methods

### 2.1. Milk

The research material was bulk milk obtained from 23 Simmental cows. The cows were kept in a traditional system on a farm located in south-eastern Poland. Feeding in the spring and summer was based mainly on pasture forage with the addition of hay and cereal meal. Milk was collected three times in the spring/summer season. Each time the volume of milk for testing was 6 litres.

The acidity of the milk was determined, i.e., potential acidity (in Soxhlet-Henkl’s degree (°SH) and expressed as lactic acid content, taking into account that 1 °SH = 0.0225% of lactic acid) according to IDF/ISO [22] and active acidity (pH) with a CP-401 pH meter (Elmetron, Zabrze, Poland). The proximate chemical composition, i.e., crude protein, fat, lactose and dry matter content, were determined with an Infrared Milk Analyzer (Bentley Instruments, Chaska, MN, USA), and casein content according to AOAC 998.06 [23]. To assess the hygienic quality of the milk, the somatic cell count (SCC) was determined by flow cytometry (Somacount 150; Bentley Instruments, Chaska, MN, USA), as well as the total microbial count (TMC) in CFU/mL by the plate method, using deep inoculation according to PN-EN ISO 8261:2002 [24] and PN-EN ISO 4833-2:2013 [25].

### 2.2. Yoghurt

The milk was subjected to heat treatment (pasteurization) at 85 °C for 30 min in a water bath. Then it was immediately cooled to 40 °C and inoculated with 0.15 g/L of one of two two freeze-dried DVS bacterial cultures intended for yoghurt production [12]. These were thermophilic yoghurt cultures from Chr. Hansen (Graasten, Denmark), i.e., Mild 1.0 Yo-Flex (hereafter Mild 1.0) and YC-X11 Yo-Flex (hereafter YC-X11), containing *Streptococcus thermophilus* and *Lactobacillus delbrueckii* subsp. *bulgaricus*. WPC (80% whey protein concentrate powder, Spomlek Dairy Cooperation, Radzyń Podlaski, Poland) was used as an additive after heat treatment and cooling down to 40 °C. The experimental design is presented in Table 1.

The inoculated milk was incubated in polypropylene (PP) plastic 100 mL containers at 40 °C (according to the manufacturer’s instructions) until pH = 4.6 was attained. The products were then cooled to 20 °C to discontinue the incubation. The yoghurts were stored at 4–6 °C until the next day (approximately 16 h) for analysis. A total of 60 yoghurts were analysed: 20 without WPC (control) and 40 with the addition of WPC.

#### Yoghurt Analysis

The active acidity (pH value) was measured before, during and after fermentation using a CP-401 pH-meter (Elmetron, Zabrze, Poland). The potential acidity was determined by the titration method (according to IDF/ISO [21]) and expressed as lactic acid content.

The water activity in the yoghurts was measured using a HygroLab C1 water activity meter (Rotronic, Bassersdorf, Switzerland). Measurements were made using the AWQ mode and stabilization for 15 min after the yoghurts had reached room temperature. The determinations were made in triplicate.

The yoghurts were analysed for content of protein (Kjeldahl method according to PN-EN ISO 8968-1:2014 [26]), fat (van Gulik’s method), and dry matter (oven-drying at 102 °C) [27]. The measurements were taken in triplicate.

The water-holding capacity (WHC), reflecting the degree of syneresis, was also determined—using the method developed by Bong and Moraru [28]. 10 g of yoghurt were weighed into a test tube and then centrifuged in a laboratory centrifuge (UNIVERSAL 320; Hettich, Tuttlingen, Germany) at 5 °C for 10 min at 1250× *g*. After the indicated time, the precipitated whey was weighed. The tests were carried out in triplicate. WHC was calculated based on the formula:WHC (%) = (10 − W)/10 × 100%,(1)
where: W—mass of the separated whey (g).

The texture parameters of the yoghurt curds were measured using a BDO-FB0.5TS universal testing machine (Zwick GmbH and Co, Ulm, Germany). The yoghurt curds were prepared in a dedicated beaker (50 mm in diameter and 150 mm high). For each yoghurt curd, two yoghurt samples were prepared and tested after approximately 20 h of storage at 4–6 °C. A beaker with the sample was centrally placed under the plunger of the apparatus with a cylindrical die 45 mm in diameter and 5 mm in height, and then compressed to a depth of 25 mm at a speed of 1 mm/s. On the basis of the force-time curves obtained, the following texture characteristics were determined for the curds: firmness (the maximum positive force, N), consistency (the positive area of the curve up to the maximum point, mJ) and cohesive strength (the maximum negative force, N). The measurements were taken in duplicate. The measurements and results were analysed using testXpert II dedicated software (Zwick/Roell, Ulm, Germany).

The colours of the samples were measured using a Minolta CR-310 Chroma Meter (Minolta Camera Co. Ltd., Osaka, Japan) using D65 as the standard light source. The milk samples were poured into small disposable Petri dishes (60 mm in diameter and a milk layer height of 10 mm), and then placed on a white standard plate. The reflectance of the milk surface was measured using a measuring head (50 mm aperture diameter; geometry 0°). The CIE colour parameters were the L* (lightness), a* (redness/greenness) and b* (yellowness/blueness) [29]. The colour tests were performed in four replications. The white index (WI) and yellowing index (YI) were calculated using the following equations [30]:WI = [(ΔL*)^2^ + (Δa*)^2^ + (Δb*)^2^]^0.5^(2)
YI = 142.86b*·L*^−1^.(3)

A sensory evaluation of the yoghurts was performed by a suitably prepared 20-person panel. Prior to the evaluation, the samples were coded and left to stand for 1 h at room temperature to reach a suitable temperature for eating, and then they were presented to the testers together with a questionnaire. A five-point scale was used to evaluate the colour, consistency, flavour and aroma of the products, in which 1 designated a disqualifying quality (unsuitable) and 5 indicated a very good quality (natural, characteristic for the product). A detailed assessment of the flavour and aroma was divided into the following qualities: yoghurt-like, milky, foreign, sour, bitter, and overall intensity. For this purpose, a six-point scale was used as follows: 0—imperceptible/none, 1—very weak, 2—weak, 3—average, 4—strong and 5—very strong. The same scale was used to evaluate the aftertaste (intensity and duration). In the case of syneresis, 0 designated absence and 1 indicated presence. The overall rating was over a range from 1 (bad) to 10 (very good) [31].

### 2.3. Statistical Analysis

A statistical analysis of the results was performed using StatSoft Inc. Statistica ver. 13.1 (Dell, Round Rock, TX, USA), using one-way (for the following parameters: acidity, proximate chemical composition and hygienic quality of bulk milk used for yoghurt production) and multi-way (for: active acidity, lactic acid content, proximate chemical composition, water activity, water-holding capacity, texture and colour parameters, and sensory evaluation of the yoghurts) analyses of variance (ANOVA). The significance of the differences between the means for the groups was determined by Mann-Whitney test at a level of *p* (alpha) = 0.05. The results are presented as the means ± standard deviation (SD).

## 3. Results and Discussion

### 3.1. Milk

One of the main determinants of the value of dairy products is the quality of raw milk. Only raw milk of adequate hygienic quality and a suitable chemical composition will provide a palatable product with a long shelf-life and optimal nutritional value and health-promoting properties, which will fully meet consumer expectations. Table 2 presents the physicochemical and hygienic quality of the bulk milk used to make the yoghurt. According to *Codex Alimentarius* [32], titratable acidity and microbiological and chemical criteria should be used to determine unacceptable conditions in milk products. The titrimetric acidity expresses the degree of buffering in milk and indicates any changes taking place in acidic compound concentrations in dairy products, even if the pH remains unchanged [33]. The pH of the milk was 6.73, which according to Król et al. [13] indicates that raw milk is fresh and suitable for processing. The milk contained 13.84% dry matter, including 4.76% fat, 3.74% total protein, and 4.73% lactose. An important component of milk, determining its suitability for processing, is casein. The milk contained 2.90% of this protein, which constituted 77.54% of the total protein. Research by other authors [34,35] on the chemical composition of raw milk obtained from cows of the same breed, i.e., Simmental, showed similar values. The hygienic quality of the raw milk used for yoghurt production was satisfactory. The somatic cell count and the total microbial count were significantly lower than the limits imposed by Commission Regulation (EC) no. 1662/2006 [36], i.e., 400,000/mL and 100,000 CFU/mL, respectively.

### 3.2. Yoghurts

#### 3.2.1. Acidity

The active acidity of the experimental yoghurts was standard: about 4.60 at the start of the study, i.e., the first day after production—day 0 of storage (Table 3). The addition of whey protein, which have buffering properties, did not significantly affect the acidity of the yoghurt. Acidity increased during the storage period, i.e., pH was lower on day 28 than on day 0, which was confirmed statistically in all cases (irrespective of the type of culture and the presence or absence of WPC). The type of culture used had a statistically significant influence (*p* ≤ 0.05 and *p* ≤ 0.01) on the acidity of yoghurt with 2% WPC on days 7, 14 and 21 of storage. Overall, the decrease in pH between the first and last day of storage was comparable (a decrease by an average 0.18 and 0.21, respectively) for yoghurts produced using the Mild 1.0 and YC-X11 strains and for all WPC variants. The decrease in active acidity during storage is consistent with the research of other authors [37,38], in which the pH range was within similar limits as in the present study, significantly exceeding 4.0. According to the authors, this decrease is explained by the increase in lactic acid content during lactic fermentation.

As a result of sugar fermentation, strains of lactic acid bacteria used in yoghurt production produce lactic acid in the amount of 0.6% to 1.0%, as well as acetaldehyde, acetoin and diacetyl, giving yoghurts specific sensory characteristics and a long shelf-life. The titrimetric acidity of the experimental yoghurts was within the normal range [27], according to which titrimetric acidity expressed as lactic acid content should be at least 0.6% (Table 4). The highest lactic acid content was found in the yoghurt made using YC-X11 strains with 2% added whey protein concentrate on day 21 (1.269%), and the lowest in the control yoghurt (without added protein) made with Mild 1.0 strains on day 0 (0.626%). The addition of whey proteins increases the buffering properties of milk, which influences the potential acidity [38].

The content of lactic acid in the yoghurts increased with the passage of storage time, which was confirmed statistically (at *p* ≤ 0.05 and *p* ≤ 0.01). The smallest increases in lactic acid content over 28 days were observed in the control yoghurts (0%) and those with 1% WPC, irrespective of the strains used. The increase in lactic acid content in yoghurts and the decrease in pH during storage are caused by the fermentation activity of the bacteria in the culture. Bacteria continue to degrade lactose under refrigerated conditions, although much more slowly than at the optimum temperature for thermophilic bacteria [39]. The results of our research regarding the amount of lactic acid are in agreement with those presented by Sady et al. [37] and Moneim et al. [40]. Sady et al. [37] found that the addition of whey protein concentrate inhibited the increase in lactic acid content, which has a major impact on product stability during refrigerated storage. Akalin et al. [41] showed no relationship between fortification of yoghurt with WPC and lactic acid increase during storage. There was also no such relationship found in our research, as both the smallest difference and the largest difference in lactic acid content between the first and the last day of storage were obtained in yoghurts with added WPC, made with the Mild 1.0 culture (with 1% WPC—0.148%; with 2% WPC—0.361%; Table 4). The interactions of the examined factors, i.e., WPC content and day of storage, proved to be significant at *p* ≤ 0.05 for lactic acid content, however, the interactions of WPC content, day of storage and type of bacterial cultures did not. A significant effect of the addition of proteins was noted only on the 28th day of storage for the Mild 1.0 yoghurts (at *p* ≤ 0.05) and on days 21 and 28 day for the YC-X11 yoghurts (*p* ≤ 0.01 and *p* ≤ 0.05, respectively). Lactic acid content was shown to depend on the type of strains used only for yoghurts with 1% WPC on day 14 (*p* ≤ 0.05) and with 2% WPC on day 28 (*p* ≤ 0.01). Many other researchers [39,42] have also shown a significant effect of the type of culture used for yoghurt production on its titratable acidity. Cultures containing *Lactobacillus bulgaricus* subsp. *delbrueckii* are believed to contribute to a greater extent to the acidity of yoghurts.

#### 3.2.2. Chemical Composition

The high quality of the raw milk translated into the high quality of the products obtained. Table 5 presents the chemical composition of the yoghurts in relation to the type of bacterial cultures used and the percentage of WPC. Yoghurts with 2% WPC contained 5.94% total protein on average, i.e., 1.72 pp more than the control yoghurts, irrespective of the type of bacterial culture (statistically significant difference at *p* ≤ 0.01). Their average fat content was 4.57%. It should be noted that fat content was not statistically affected by the type of bacterial culture used or the addition of WPC. According to Brodziak et al. [16], the addition of whey protein in the form of WPC to yoghurt increases the content of non-fat and total dry matter, which was confirmed by the present study. This is due not only to an increase in the total protein content, but to increased lactose content as well (not determined in this study, however, according to the manufacturer’s declaration on the package WPC contained 8% of lactose). 

The yoghurts with added WPC contained more dry matter than those without whey proteins, although the differences were significant only between the control group and the yoghurts with 2% WPC made with the Mild 1.0 culture (Table 5). Regarding the storage period, significant differences (at *p* ≤ 0.05) were found in the content of non-fat and total dry matter between the start (day 0) and the end (28 day) of the refrigeration period.

#### 3.2.3. Water-Holding Capacity

The water-holding capacity (WHC) of yoghurts is an indicator of their ability to retain serum (whey) in the gel structure [43]. Separation of whey from yoghurt is a natural phenomenon, although it is negatively perceived by consumers, who generally associate it with adverse changes in quality and as a sign of deterioration. For this reason yoghurts with a low degree of syneresis are preferred by consumers. Additives can be used to achieve this purpose. The water-holding capacity of the yoghurts during storage, indicating the amount of syneresis, is shown in Table 6. The results indicate the effect of all three factors included in the study. On the first day of analysis, yoghurts without added whey protein, irrespective of the type of strain, had a significantly (*p* ≤ 0.01) higher water-holding capacity than the products containing 1% and 2% WPC. For yoghurts made with Mild 1.0, this difference was greater, amounting to 15.7% (20 pp). However, from day 7 of storage the trend reversed and persisted until the end of the study. Yoghurts containing WPC attained and maintained higher water-holding capacity than the control yoghurts, although this was statistically confirmed only on the 7th day of storage. The type of bacterial culture used was also found to affect WHC. Throughout the study period, there was a statistically significant difference (*p* ≤ 0.01 on day 0 and *p* ≤ 0.05 on other days) between Mild 1.0 and YC-X11 for yoghurts containing 2% WPC. Amatayakul et al. [44] also reported that the amount of syneresis depends on the type of bacterial culture used. Overall, WHC decreased significantly over storage time, reaching its lowest values on the 28th day of storage (Table 6). 

This clearly indicates an increase in syneresis in all tested yoghurts. The largest difference (by approx. 30 pp) in water-holding capacity between the initial and final day of storage was found for yoghurts without WPC, i.e., 26.33% for Mild 1.0 and 28.13% for YC-X11. These trends are supported by Sodini et al. [8], who showed that the water-holding capacity is greater in milk-based beverages with added whey protein than without it. Kozioł et al. [45] state that higher water-holding capacity in products containing WPC may result from increased cross-linking of curds compared to yoghurts not enriched with protein preparations. According to Gharibzahedi and Chronakis [46], the three-dimensional structure of yoghurt is stabilized by the reduction of syneresis resulting from cross-linking of microbial transglutaminase with milk proteins. Whey separation (syneresis) therefore depends on the capacity of proteins to retain water, which in turn depends on the protein content of the product and the type of protein [47]. Lee and Lucey [43] showed that dry matter content and the ratio of casein to whey proteins are important factors influencing the water-holding capacity of yoghurt curd. According to Puvanenthiran et al. [48], a decrease in the casein-to-whey protein ratio in yoghurt made from milk with WPC, relative to yoghurt made from reconstituted milk, led to increased gel firmness. Akalin et al. [41] also showed that WPC enhanced water-holding capacity more than caseinate. The authors observed that the control yoghurts exhibited the highest level of syneresis throughout the 28 days of storage, whereas low-level syneresis was obtained by fortification with WPC at d 14 and 28 (*p* < 0.05). Lower water-holding capacity and higher syneresis values were reported in the yoghurt with added caseinate. In practice, additives are sought to improve water retention by the yoghurt curd.

#### 3.2.4. Water Activity

The water activity (a_w_) parameter can be used to determine the course of biochemical reactions, the stability of the sensory characteristics of food, the development of microorganisms, and above all the storage stability of food products. Unfavourable reactions affecting food quality are more dependent on the state of the water than on its content in the product [49]. Water activity (a_w_) is the availability of the water contained in the product for microbes. The higher the a_w_ index, the faster microorganisms can multiply, using the water for their own processes. The a_w_ value can be controlled by thickening, drying, and regulating the osmotic pressure of products. In the case of dairy products, it is also possible to regulate the pH of the product to values that will limit the development of unfavourable microbes without affecting beneficial bacterial cultures [50]. Table 7 shows the changes in water activity in the products over time. The type of strains used had no significant impact on the changes in this parameter. The effect of the addition of milk proteins (except for yoghurts made with YC-X11 on the 7th and 14th day of storage) was not statistically confirmed, but in most cases, irrespective of the type of strain, it decreased as WPC content increased. However, irrespective of the type of strain and the addition of WPC, a_w_ was found to increase significantly with the storage time. Greater changes were observed in the yoghurts made using Mild 1.0 bacteria.

#### 3.2.5. Texture

The choice of yoghurt production method is important for the texture of the final product. It should be noted that in the water bath method, once the fermented beverage is obtained it is no longer possible to interfere with the contents of the package. For this reason, this method is usually used to produce plain fermented beverages, i.e., without flavouring additives. According to Lucey et al. [51], the textural properties of products largely depend on their protein content. Glibowski and Rybak [12], and Nishinari et al. [52] report that the texture of yoghurt is also influenced by the addition of powdered milk, whey proteins, polysaccharides, inulin, or additional thickeners, as well as by the homogenization process. Therefore, changes in texture parameters should also have been expected in our research. For consumers, firmness is one of the most important rheological properties and has a major impact on purchasing decisions [16]. The changes in the firmness of the experimental yoghurts over 28 days of storage are shown in Table 8. The highest value for firmness was found for yoghurts without added whey protein at the start of storage (day 0)—6.21 (Mild 1.0) and 6.32 N (YC-X11). The addition of WPC significantly affected the firmness of yoghurts made with both strains, with the lowest values noted in most cases in the products with 2% WPC. The greatest difference (4.25 N) between the control yoghurt and yoghurt with 2% WPC was obtained for the Mild 1.0 culture on day 28 of storage. This sample had the lowest curd firmness—1.70 N (*p* ≤ 0.01). Research by Glibowski and Krępacka [53] confirms that as the concentration of whey protein increases, the firmness of yoghurt decreases. In our research, this parameter also decreased with the passage of storage time, and the changes in all cases were statistically significant. The results of our research differ from those of Gustaw [54], who found that the firmness of gels produced using WPC increased with storage time. Mituniewicz-Małek et al. [55] reported that firmness depends on the type of bacterial culture used, which was also demonstrated in our study. They also found that the rheological features of dairy products depend on their pH, i.e., the lower the pH, the greater the firmness, but this was not confirmed by our results.

Storage time significantly (*p* ≤ 0.01) reduced consistency in all yoghurt samples, as shown in Table 9. The largest decrease in consistency during refrigerated storage was observed for yoghurts containing 2% WPC. The difference in this parameter between the first and final day of storage was 3.73 pp in products made with the Mild 1.0 culture and as much as 4.96 pp for YC-X11. Pawlos et al. [56] reported a similar tendency (2014). The consistency of the system is weaker the closer the consistency value is to zero [57]. In our research, the amount of added whey protein significantly influenced the consistency of yoghurts, both those made with Mild 1.0 and those with YC-X11. As the share of WPC increased, consistency increased, except for the last day of storage (Table 9). Analysis of the effect of the bacterial culture used revealed significant differences only on the seventh day for both amounts of WPC.

The results also indicate a significant impact of all three experimental factors on other texture characteristic, i.e., cohesive strength (Table 10). The values of cohesive strength for yoghurts significantly (*p* ≤ 0.01) decreased with the addition of WPC. Over time in storage, the cohesive strength of yoghurts, both with and without added WPC, also significantly decreased. Moreover, it should be noted that the type of bacterial culture influenced this parameter throughout the storage period. Generally, the yoghurt curds with WPC were more strength as reported other authors [30,58].

#### 3.2.6. Colour Parameters

As pointed out by Mituniewicz-Małek et al. [59], colour is the first characteristic to determine a product’s sensory appeal. The CIELab system (L*a*b*) is currently the most common means of describing colour and is the basis of modern colour management systems. Our results indicate that all the experimental factors had a significant impact on the L* parameter (lightness), a* (a change in the green to red range), and b* (a change in the blue to yellow range)—Table 11. The highest lightness value was found for yoghurt without added WPC at the start of the storage period (95.67), and the lowest for yoghurt with 1% WPC on day 28 (89.76) made with YC-X11. The lightness of the products decreased over time. All yoghurts were the least light on the last day of the study, which was confirmed statistically (*p* ≤ 0.01). With the passage of storage time, the yoghurts became both more red (change in a*) and more yellow (parameter b*). Parameter a* was negative in all cases (from −4.56 to −0.68). In general, these values significantly (*p* ≤ 0.01) increased (approached 0) as the amount of added WPC increased. The b* parameter was positive for all yoghurt samples, ranging from 17.74 (YC-X11, 1% WPC) to 19.37 (Mild 1.0, 2% WPC). The white index (WI) of the yoghurt with added whey proteins was further from the ideal white standard than the control yoghurt, although this was not confirmed statistically until the 28th day of storage (*p* ≤ 0.05). The increased share of whey proteins therefore gave the yoghurts a less white colour. The distance from perfectly white yoghurt significantly increased with storage time in all cases, as the yoghurt became less white during storage. This feature, like the yellowing index, was not influenced by the bacterial strain used. Yoghurt with whey protein had a higher yellowing index than the control yoghurt. This change was also determined by the storage period (Table 11). The total colour difference (ΔE*) for the yoghurts ranged from 4.36 to 4.66, which indicates a marked colour deviation. The trends are consistent with results obtained by Bierzuńska et al. [30] for yoghurts made with commercial pasteurized cow milk and also containing WPC80. Rój and Przybyłowski [60] evaluated the colour of natural yoghurts with different fat content from various manufacturers. They showed that the L* parameter ranged from 87.00 to 92.58, with the lowest values obtained in the yoghurt with 0.2% fat content and the highest in the yoghurt with 10% fat. The a* values were negative (from −4.82 to −3.49), while b* had positive values ranging from 7.18 to 10.08. Cais-Sokolińska and Pikul [61] reported that the degree to which the L*, a* and b* colour parameters decreased during storage time depended on the initial pH of the yoghurts. This was true of the L* parameter in the present study. Moreover, the change in the b* parameter (an increase during storage) could have been due to destabilization of casein micelles in the pasteurization process [62].

### 3.2.7. Sensory Characteristics

Sensory evaluation, taking into account colour, consistency, flavour and aroma, is a key tool enabling assessment of consumer acceptance of a new product. All three experimental factors were found to have a highly significant (*p* ≤ 0.01) influence on the sensory features, except for the starter culture in relation to consistency and the amount of added WPC in relation to colour (Figure 1, Figure 2 and Figure 3). Testers gave the highest scores to yoghurts made using the Mild 1.0 culture without the addition of whey protein on day 0 of storage. With the passage of time, the colour, consistency, flavour and aroma changed, which was observed and negatively assessed by the tasters. This applied in particular to flavour (Figure 3). Regarding the interaction of factors, only colour (*p* ≤ 0.05) and flavour (*p* ≤ 0.01) were found to be significantly determined by the starter culture × % WPC interaction (Figure 4 and Figure 5). Akalin et al. [41] noted no significant differences between yoghurts with different amounts of WPC in terms of appearance, aroma, flavour, and overall acceptability during storage (*p* > 0.05). New products introduced to the market must gain consumer acceptance, which is not always the case with this additive. The factors playing the most important role in determining the market behaviour of consumers are the freshness and flavour of yoghurt.

In addition, a detailed assessment of the flavour and aroma was conducted, taking into account the following qualities: yoghurt-like, milky, foreign, sour, bitter, and overall intensity. All yoghurts were defined as strong yoghurt-like and milky. According to the evaluation panel, the yoghurts with added WPC were weak bitter, however, it was not statistically confirmed. Perhaps the addition was not large enough to make the difference statistically significant. Regardless of the WPC addition, aftertaste (intensity and duration) was comparable for all yoghurts. However, higher values in this regard were given to the yoghurts based on YC-X11 Yo-Flex. The use of WPC addition reduced whey leakage (syneresis) during refrigerated storage. Syneresis was visually detected in yoghurts without WPC. The overall rating for the yoghurts without WPC was 9 points (out of 10) and with the addition—7 points. However, Akalin et al. [41] stated no significant differences between the experimental yogurts with and without WPC in terms of sensory attributes.

## 4. Conclusions

To sum up, a comprehensive assessment of the physicochemical and sensory changes occurring during refrigerated storage of yoghurts manufactured with the addition of WPC and using different cultures is crucial for modelling such a product. As the cultures contained the same bacterial strains, i.e., *Streptococcus thermophilus* and *Lactobacillus delbrueckii* subsp. *bulgaricus*, these were not the primary factor differentiating the analysed features. A significant effect was found in the case of the water-holding capacity, firmness, consistency and colour parameters of the curd. The use of whey protein concentrate determined both the physicochemical and sensory quality of the yoghurts. The additive had a positive effect on potential acidity, inhibiting the increase in the amount of lactic acid in yoghurts during storage, and also reduced syneresis. However, it reduced the lightness of the curd and adversely affected sensory characteristics, mainly flavour. Moreover, nearly all parameters changed significantly (usually negatively) with the passage of storage time. The exceptions were total protein and fat content. However, these were not major changes and remained at an acceptable level. In general, the addition of 1% or 2% whey protein to yoghurt may be a good solution that can be routinely applied in the dairy industry to offer consumers a new functional product. Moreover, it offers a use for WPC, a whey product which nowadays is perceived as valuable in the dairy industry.

## Figures and Tables

**Figure 1 animals-10-01350-f001:**
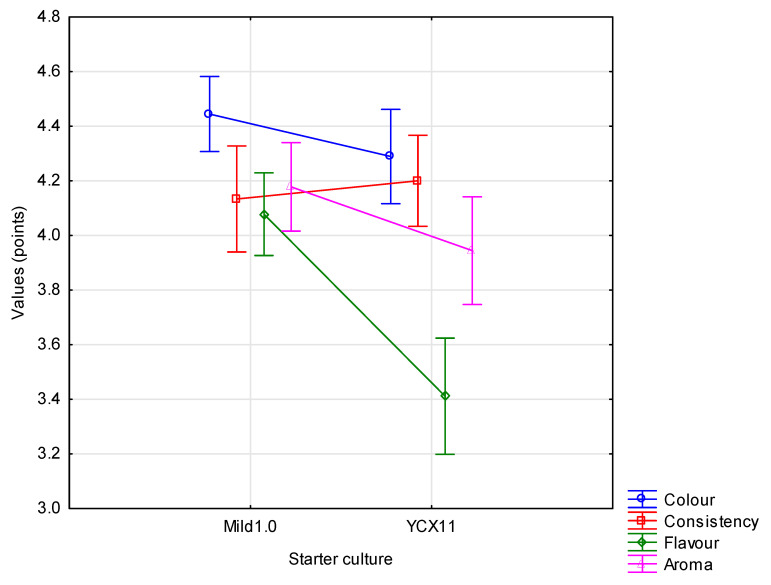
Organoleptic characteristics of yoghurts depending on the starter culture ($\bar{x}\;$ ± SD).

**Figure 2 animals-10-01350-f002:**
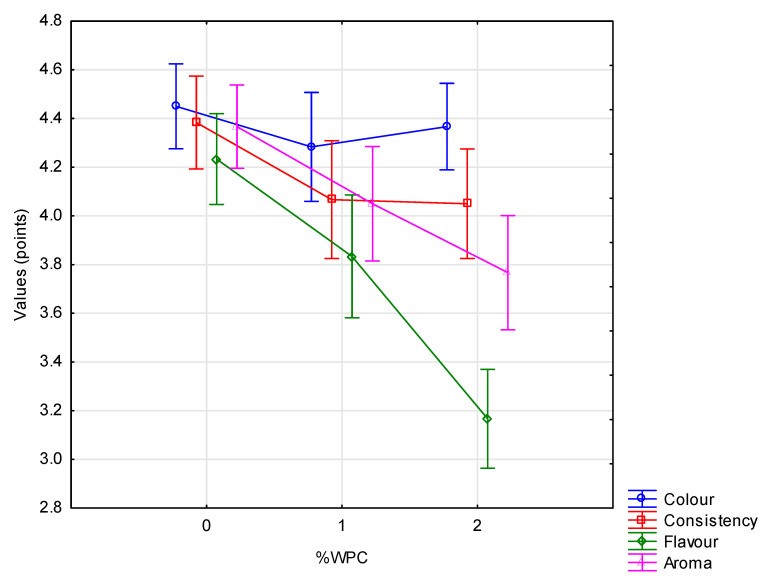
Organoleptic characteristics of yoghurts depending on the WPC content ($\bar{x}\;$ ± SD).

**Figure 3 animals-10-01350-f003:**
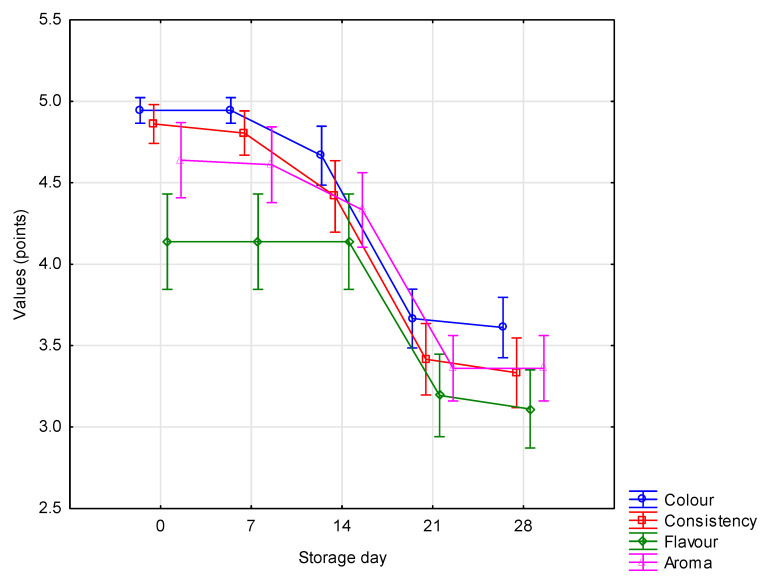
Organoleptic characteristics of yoghurts depending on the storage time ($\bar{x}\;$ ± SD).

**Figure 4 animals-10-01350-f004:**
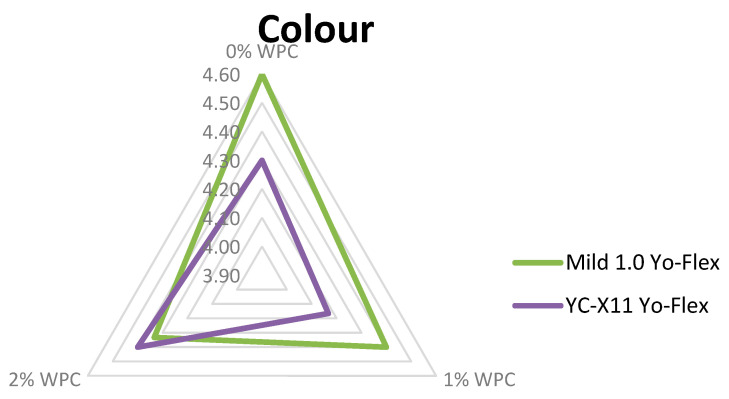
Colour of yoghurt in relation to the type of bacterial culture and the WPC content.

**Figure 5 animals-10-01350-f005:**
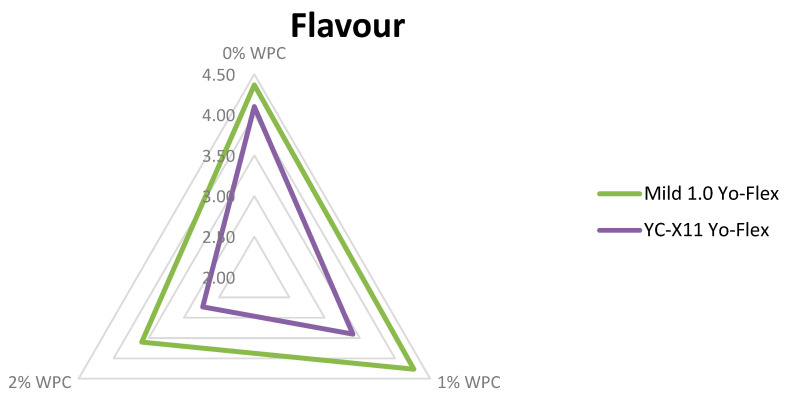
Flavour of yoghurt in relation to the type of bacterial culture and the WPC content.

**Table 1 animals-10-01350-t001:** Design of the experiment—type of bacterial cultures and percentage of WPC in yoghurts.

Mild 1.0 Yo-Flex	YC-X11 Yo-Flex
WPC content (%)	
0	0
1	1
2	2

**Table 2 animals-10-01350-t002:** Acidity, proximate chemical composition and hygienic quality of bulk milk used for yoghurt production ($\bar{x}\;$ ± SD).

Acidity		Proximate Chemical Composition						Hygienic Quality	
active (pH value)	Potential-lactic acid content (%)	Fat (%)	Protein (%)	Casein (%)	Lactose (%)	Non-fat dry matter (%)	Dry matter (%)	SCC (thous./mL)	TMC (thous. CFU/mL)
6.73 ± 0.03	0.155 ± 0.004	4.76 ± 0.06	3.74 ± 0.04	2.90 ± 0.07	4.73 ± 0.05	9.02 ± 0.07	13.84 ± 0.05	203 ± 37	6.7 × 10^4^

SCC—somatic cell count. TMC—total microbial count.

**Table 3 animals-10-01350-t003:** Active acidity (pH value) of yoghurts in relation to the type of bacterial cultures used, WPC content and storage time ($\bar{x}\;$ ± SD).

Type of Bacterial Cultures	WPC Content (%)	Storage Time (Days)				
		0	7	14	21	28
**Mild 1.0 Yo-Flex**	0	4.62 ^y^ ± 0.07	4.61 ^y^ ± 0.03	4.56 ^y^ ± 0.06	4.49 ^x^ ± 0.05	4.38 ^w^ ± 0.15
	1	4.56 ^x^ ± 0.06	4.57 ^x^ ± 0.05	4.55 ^x^ ± 0.14	4.56 ^x^ ± 0.09	4.41 ^w^ ± 0.07
	2	4.54 ^X^ ± 0.08	4.58 ^X**^ ± 0.05	4.56 ^X*^ ± 0.10	4.55 ^X**^ ± 0.04	4.39 ^W^ ± 0.09
**YC-X11 Yo-Flex**	0	4.55 ^y^ ± 0.08	4.47 ^x^ ± 0.08	4.49 ^x^ ± 0.07	4.42 ^x^ ± 0.11	4.34 ^w^ ± 0.13
	1	4.53 ^Y^ ± 0.07	4.57 ^Y^ ± 0.07	4.42 ^X^ ± 0.05	4.38 ^WX^ ± 0.12	4.30 ^W^ ± 0.19
	2	4.52 ^y^ ± 0.05	4.39 ^wx**^ ± 0.10	4.44 ^x*^ ± 0.14	4.37 ^w**^ ± 0.06	4.32 ^w^ ± 0.06

^w^, ^x^, ^y^, ^W^, ^X^, ^Y^—differences between days of storage; ^w^, ^x^, ^y^—significant differences at *p* ≤ 0.05; ^W^, ^X^, ^Y^—significant differences at *p* ≤ 0.01. *, **—differences between bacterial cultures used within a percentage of WPC; *—significant differences at *p* ≤ 0.05; **—significant differences at *p* ≤ 0.01.

**Table 4 animals-10-01350-t004:** Lactic acid content (%) in yoghurts in relation to the type of bacterial cultures used, WPC content and storage time ($\bar{x}\;$ ± SD).

Type of Bacterial Cultures	WPC Content (%)	Storage Time (Days)				
		0	7	14	21	28
**Mild 1.0 Yo-Flex**	0	0.626 ^W^ ± 0.039	0.810 ^X^ ± 0.029	0.972 ^Z^ ± 0.043	0.918 ^Y^ ± 0.046	0.819 ^aX^ ± 0.050
	1	0.698 ^W^ ± 0.051	0.837 ^X^ ± 0.046	1.026 ^Y*^ ± 0.061	1.053 ^Y^ ± 0.072	0.846 ^aX^ ± 0.048
	2	0.737 ^W^ ± 0.043	0.828 ^X^ ± 0.021	1.062 ^Z^ ± 0.031	0.972 ^Y**^ ± 0.037	1.098 ^bZ^ ± 0.061
**YC-X11 Yo-Flex**	0	0.684 ^w^ ± 0.020	0.765 ^w^ ± 0.040	0.855 ^x^ ± 0.049	0.896 ^Ax^ ± 0.031	0.855 ^ax^ ± 0.035
	1	0.676 ^W^ ± 0.026	0.711 ^W^ ± 0.018	0.846 ^X*^ ± 0.025	1.026 ^BY^ ± 0.048	0.864 ^aX^ ± 0.028
	2	0.729 ^W^ ± 0.057	0.909 ^X^ ± 0.034	0.936 ^X^ ± 0.029	1.269 ^CZ**^ ± 0.075	1.035 ^bY^ ± 0.054

^a^, ^b^, ^A^, ^B^, ^C^—differences between the percentage of WPC within a bacterial culture; ^a^, ^b^—significant differences at *p* ≤ 0.05; ^A^, ^B^, ^C^—significant differences at *p* ≤ 0.01. ^w^, ^x^, ^W^, ^X^, ^Y^, ^Z^—differences between days of storage; ^w^, ^x^—significant differences at *p* ≤ 0.05; ^W^, ^X^, ^Y^, ^Z^—significant differences at *p* ≤ 0.01. *, **—differences between bacterial cultures used within a percentage of WPC; *—significant differences at *p* ≤ 0.05; **—significant differences at *p* ≤ 0.01.

**Table 5 animals-10-01350-t005:** Proximate chemical composition of yoghurts in relation to the type of bacterial cultures used, WPC content and storage time ($\bar{x}\;$ ± SD).

Type of Bacterial Cultures	WPC Content (%)	Protein (%)		Fat (%)		Non-Fat Dry Matter (%)		Dry Matter (%)	
Storage time (day)		0	28	0	28	0	28	0	28
Mild 1.0 Yo-Flex	0	4.19 ^A^ ± 0.10	4.16 ^A^ ± 0.14	4.54 ± 0.14	4.54 ± 0.19	8.07 ^ax^ ± 0.16	7.64 ^aw^ ± 0.10	12.64 ^ax^ ± 0.12	12.07 ^aw^ ± 0.20
	1	5.06 ^B^ ± 0.19	5.02 ^B^ ± 0.21	4.60 ± 0.15	4.60 ± 0.11	8.95 ^bx^ ± 0.10	8.42 ^bw^ ± 0.08	13.59 ^bx^ ± 0.18	13.04 ^bw^ ± 0.13
	2	5.91 ^C^ ± 0.14	5.87 ^C^ ± 0.12	4.59 ± 0.18	4.53 ± 0.22	10.01 ^cx^ ± 0.14	9.67 ^cw^ ± 0.16	14.62 ^cx^ ± 0.21	14.05 ^cw^ ± 0.21
YC-X11 Yo-Flex	0	4.26 ^A^ ± 0.15	4.22 ^A^ ± 0.16	4.56 ± 0.13	4.53 ± 0.14	8.02 ^ax^ ± 0.12	7.52 ^aw^ ± 0.12	12.58 ^ax^ ± 0.10	12.10 ^aw^ ± 0.09
	1	5.14 ^B^ ± 0.16	5.12 ^B^ ± 0.18	4.54 ± 0.15	4.53 ± 0.20	8.99 ^bx^ ± 0.17	8.48 ^bw^ ± 0.15	13.55 ^bx^ ± 0.16	12.98 ^bw^ ± 0.18
	2	5.97 ^C^ ± 0.13	5.93 ^C^ ± 0.15	4.59 ± 0.16	4.56 ± 0.10	10.06 ^cx^ ± 0.09	9.46 ^cw^ ± 0.11	14.60 ^cx^ ± 0.14	13.95 ^cw^ ± 0.24

^a^, ^b^, ^c^, ^A^, ^B^, ^C^—differences between the percentage of WPC within a bacterial culture; ^a^, ^b^, ^c^—significant differences at *p* ≤ 0.05; ^A^, ^B^, ^C^—significant differences at *p* ≤ 0.01. ^w^, ^x^—differences between days of storage within the component, significant at *p* ≤ 0.05.

**Table 6 animals-10-01350-t006:** Water holding capacity—WHC (%, determined at 5 °C) of yoghurts in relation to the type of bacterial cultures used, WPC content and storage time ($\bar{x}\;$ ± SD).

Type of Bacterial Cultures	WPC Content (%)	Storage Time (day)				
		0	7	14	21	28
Mild 1.0 Yo-Flex	0	80.35 ^BY^ ± 2.99	57.90 ^aW^ ± 4.52	59.65 ^W^ ± 8.49	66.18 ^X^ ± 3.36	54.02 ^W*^ ± 1.58
	1	64.57 ^Ax**^ ± 1.05	66.89 ^bx^ ± 1.05	62.55 ^x^ ± 3.06	67.05 ^x*^ ± 2.18	57.13 ^w^ ± 4.25
	2	64.73 ^Ax**^ ± 1.95	64.33 ^abx*^ ± 2.58	61.68 ^w*^ ± 3.07	69.65 ^x*^ ± 1.30	56.33 ^w*^ ± 3.97
YC-X11 Yo-Flex	0	79.56 ^By^ ± 2.09	57.48 ^aw^ ± 4.28	59.70 ^w^ ± 3.29	65.72 ^x^ ± 3.83	51.43 ^aw*^ ± 1.20
	1	76.75 ^BY**^ ± 3.08	65.48 ^bW^ ± 5.41	60.68 ^W^ ± 4.43	71.23 ^X*^ ± 1.60	59.12 ^bW^ ± 3.88
	2	69.16 ^Ax**^ ± 2.10	61.18 ^abx*^ ± 4.70	58.62 ^w*^ ± 2.26	66.30 ^x*^ ± 3.56	53.17 ^aw*^ ± 1.45

^a^, ^b^, ^A^, ^B^—differences between the percentage of WPC within a bacterial culture; ^a^, ^b^—significant differences at *p* ≤ 0.05; ^A^, ^B^—significant differences at *p* ≤ 0.01. ^w^, ^x^, ^y^, ^W^, ^X^, ^Y^—differences between days of storage; ^w^, ^x^, ^y^—significant differences at *p* ≤ 0.05; ^W^, ^X^, ^Y^—significant differences at *p* ≤ 0.01. *, **—differences between bacterial cultures used within a percentage of WPC; *—significant differences at *p* ≤ 0.05; **—significant differences at *p* ≤ 0.01.

**Table 7 animals-10-01350-t007:** Water activity (a_w_) of yoghurts in relation to the type of bacterial cultures used, WPC content and storage time ($\bar{x}$ ± SD).

Type of Bacterial Cultures	WPC Content (%)	Storage Time (Days)				
		0	7	14	21	28
Mild 1.0 Yo-Flex	0	0.938 ^W^ ± 0.002	0.948 ^W^ ± 0.004	0.953 ^X^ ± 0.017	0.964 ^Y^ ± 0.013	0.966 ^Y^ ± 0.012
	1	0.934 ^W^ ± 0.003	0.946 ^X^ ± 0.006	0.956 ^X^ ± 0.009	0.950 ^X^ ± 0.020	0.968 ^Y^ ± 0.011
	2	0.938 ^W^ ± 0.002	0.952 ^X^ ± 0.003	0.956 ^X^ ± 0.013	0.961 ^XY^ ± 0.017	0.968 ^Y^ ± 0.013
YC-X11 Yo-Flex	0	0.958 ^X^ ± 0.002	0.962 ^bX^ ± 0.002	0.966 ^bX^ ± 0.002	0.947 ^W^ ± 0.001	0.971 ^X^ ± 0.008
	1	0.956 ^x^ ± 0.004	0.958 ^abx^ ± 0.002	0.948 ^aw^ ± 0.008	0.948 ^w^ ± 0.001	0.960 ^x^ ± 0.010
	2	0.950 ^W^ ± 0.002	0.952 ^aW^ ± 0.005	0.954 ^abWX^ ± 0.003	0.948 ^W^ ± 0.002	0.962 ^X^ ± 0.007

^a^, ^b^—differences between the percentage of WPC within a bacterial culture significant at *p* ≤ 0.05. ^w^, ^x^, ^W^, ^X^, ^Y^—differences between days of storage; ^w^, ^x^—significant differences at *p* ≤ 0.05; ^W^, ^X^, ^Y^—significant differences at *p* ≤ 0.01.

**Table 8 animals-10-01350-t008:** Firmness (N) of yoghurts in relation to the type of bacterial cultures used, WPC content and storage time ($\bar{x}\;$ ± SD).

Type of Bacterial Cultures	WPC Content (%)	Storage Time (Days)				
		0	7	14	21	28
Mild 1.0 Yo-Flex	0	6.21 ^z^ ± 1.50	3.37 ^aw**^ ± 0.85	4.78 ^bx **^ ± 0.98	5.76 ^By**^ ± 1.45	5.95 ^By*^ ± 1.69
	1	6.17 ^Z^ ± 1.73	4.51 ^abX*^ ± 0.49	3.80 ^abW^ ± 0.77	4.97 ^BXY^ ± 1.14	5.09 ^BY**^ ± 1.75
	2	5.36 ^Y^ ± 1.63	5.11 ^bY^ ± 0.73	2.46 ^bX*^ ± 0.51	2.25 ^AWX*^ ± 0.97	1.70 ^AW**^ ± 0.58
YC-X11 Yo-Flex	0	6.32 ^By^ ± 1.34	6.14 ^cY**^ ± 0.92	2.26 ^aW **^ ± 0.34	3.04 ^W**^ ± 1.28	4.90 ^X*^ ± 1.62
	1	5.32 ^aby^ ± 1.28	5.79 ^by*^ ± 1.31	4.18 ^bx^ ± 1.13	3.97 ^wx^ ± 1.40	3.35 ^w**^ ± 1.07
	2	4.47 ^ax^ ± 1.57	4.54^ax^ ± 0.96	3.98 ^bw*^ ± 0.99	3.94 ^w*^ ± 1.17	3.91 ^w**^ ± 1.39

^a^, ^b^, ^c^, ^A^, ^B^—differences between the percentage of WPC within a bacterial culture; ^a^, ^b^, ^c^—significant differences at *p* ≤ 0.05; ^A^, ^B^—significant differences at *p* ≤ 0.01. ^w^, ^x^, ^y^, ^z^, ^W^, ^X^, ^Y^—differences between days of storage; ^w^, ^x^, ^y^, ^z^—significant differences at *p* ≤ 0.05; ^W^, ^X^, ^Y^—significant dferences at *p* ≤ 0.01. *, **—differences between bacterial cultures used within a percentage of WPC; *—significant differences at *p* ≤ 0.05; **—significant differences at *p* ≤ 0.01.

**Table 9 animals-10-01350-t009:** Consistency (mJ) of yoghurts in relation to the type of bacterial cultures used, WPC content and storage time ($\bar{x}\;$ ± SD).

Type of Bacterial Cultures	WPC Content (%)	Storage Time (Days)				
		0	7	14	21	28
Mild 1.0 Yo-Flex	0	3.78 ^ay^ ± 1.03	1.95 ^Aw**^ ± 0.46	2.02 ^aw^ ± 0.20	2.25 ^w^ ± 0.69	2.74 ^bx^ ± 0.71
	1	4.47 ^abX^ ± 1.67	5.34 ^CX*^ ± 1.30	2.24 ^aW*^ ± 0.76	2.38 ^W*^ ± 0.88	2.17 ^bW^ ± 0.53
	2	5.21 ^By^ ± 1.24	3.29 ^BX*^ ± 0.32	3.37 ^bX^ ± 0.96	3.00 ^X^ ± 0.24	1.48 ^aW^ ± 0.38
YC-X11 Yo-Flex	0	2.93 ^AX^ ± 0.64	3.90 ^y**^ ± 1.12	1.54 ^AW^ ± 0.17	1.73 ^AW^ ± 0.64	1.90 ^AW^ ± 0.72
	1	5.02 ^BY^ ± 1.25	3.98 ^X*^ ± 0.37	3.79 ^BX*^ ± 0.58	3.47 ^BX*^ ± 0.27	2.67 ^BW^ ± 0.65
	2	6.12 ^BZ^ ± 1.19	4.75 ^YZ*^ ± 1.06	3.74 ^BY^ ± 1.14	2.81 ^ABX^ ± 0.86	1.16 ^AW^ ± 0.20

^a^, ^b^, ^A^, ^B^, ^C^—differences between the percentage of WPC within a bacterial culture; ^a^, ^b^—significant differences at *p* ≤ 0.05; ^A^, ^B^, ^C^—significant differences at *p* ≤ 0.01. ^w^, ^x^, ^y^, ^W^, ^X^, ^Y^, ^Z^—differences between days of storage; ^w^, ^x^, ^y^—significant differences at *p* ≤ 0.05; ^W^, ^X^, ^Y^, ^Z^—significant differences at *p* ≤ 0.01. *, **—differences between bacterial cultures used within a percentage of WPC; *—significant differences at *p* ≤ 0.05; **—significant differences at *p* ≤ 0.01.

**Table 10 animals-10-01350-t010:** Cohesive strength (N) of yoghurts in relation to the type of bacterial cultures used, WPC content and storage time ($\bar{x}\;$ ± SD).

Type of Bacterial Cultures	WPC Content (%)	Storage Time (Days)				
		0	7	14	21	28
Mild 1.0 Yo-Flex	0	0.96 ^Cx**^ ± 0.10	1.06 ^Cx*^ ± 0.15	0.97 ^Cx*^ ± 0.08	0.86 ^Cw**^ ± 0.10	0.78 ^Cw**^ ± 0.12
	1	0.74 ^Bx**^ ± 0.08	0.69 ^Bx**^ ± 0.11	0.79^Bx**^ ± 0.06	0.64 ^Bw**^ ± 0.09	0.59 ^Bw**^ ± 0.08
	2	0.45 ^AY**^ ± 0.06	0.43 ^AY^ ± 0.07	0.45 ^AX*^ ± 0.05	0.40 ^AWX*^ ± 0.07	0.37 ^AW**^ ± 0.08
YC-X11 Yo-Flex	0	1.22 ^Cx**^ ± 0.14	1.14 ^Cw*^ ± 0.10	1.11 ^Bw*^ ± 0.13	1.16 ^Cw**^ ± 0.10	1.12 ^Cw**^ ± 0.13
	1	1.04 ^Bx**^ ± 0.12	1.00 ^Bxw**^ ± 0.11	1.05 ^Bx**^ ± 0.09	0.97 ^Bxw**^ ± 0.07	0.93 ^Bw**^ ± 0.10
	2	0.63 ^Ay**^ ± 0.08	0.59 ^Awy^ ± 0.06	0.55 ^Axw*^ ± 0.06	0.54 ^Aw*^ ± 0.06	0.51 ^Aw**^ ± 0.07

^A^, ^B^, ^C^—differences between the percentage of WPC within a bacterial culture significant at *p* ≤ 0.01. ^w^, ^x^, ^y^—differences between days of storage significant at *p* ≤ 0.05. *, **—differences between bacterial cultures used within a percentage of WPC; *—significant differences at *p* ≤ 0.05; **—significant differences at *p* ≤ 0.01.

**Table 11 animals-10-01350-t011:** Colour parameters of yoghurts in relation to the type of bacterial cultures used, WPC content and storage time ($\bar{x}\;$ ± SD).

Type of Bacterial Cultures	WPC Content (%)	L*		a*		b*		WI		YI	
Storage Time (days)		0	28	0	28	0	28	0	28	0	28
Mild 1.0 Yo-Flex	0	95.17 ^CX*^ ± 0.15	92.91 ^CW*^ ± 0.41	−4.56 ^AW*^ ± 0.04	−0.95 ^BX**^ ± 0.07	18.54 ^AW**^ ± 0.04	18.93 ^AX*^ ± 0.24	19.68 ± 0.34	20.30 ^a^ ± 0.21	27.83 ^aW^ ± 0.52	29.12 ^AX^ ± 0.30
	1	94.10 ^AX^ ± 0.16	91.67 ^BW*^ ± 0.29	−3.44 ^BW^ ± 0.05	−2.24 ^AX**^ ± 0.10	18.50 ^AW**^ ± 0.02	18.92 ^AX**^ ± 0.26	19.72 ^w^ ± 0.23	20.79 ^ax^ ± 0.18	28.08 ^abw**^ ± 0.27	29.48 ^Ax^ ± 0.43
	2	94.68 ^BX*^ ± 0.25	90.15 ^AW^ ± 0.26	−3.12 ^CW*^ ± 0.01	−1.07 ^CX**^ ± 0.05	19.05 ^BW**^ ± 0.03	19.37 ^BX*^ ± 0.19	20.04 ^W^ ± 0.32	21.75 ^bX^ ± 0.27	28.74 ^bW^ ± 0.55	30.69 ^BX^ ± 0.61
YC-X11 Yo-Flex	0	95.67 ^CX*^ ± 0.22	91.73 ^CW*^ ± 0.28	−4.35 ^AW*^ ± 0.02	−2.96 ^AX**^ ± 0.08	18.10 ^BW**^ ± 0.02	18.59 ^BX*^ ± 0.14	19.11 ^w^ ± 0.15	20.57 ^ax^ ± 0.34	27.03 ^Aw^ ± 0.37	28.96 ^ax^ ± 0.44
	1	94.07 ^AX^ ± 0.20	89.76 ^AW*^ ± 0.36	−3.37 ^BW^ ± 0.02	−1.89 ^BX**^ ± 0.05	17.74 ^AW**^ ± 0.02	18.01 ^AX**^ ± 0.25	19.00 ^W^ ± 0.26	20.80 ^abX^ ± 0.41	26.94 ^AW**^ ± 0.26	28.68 ^aX^ ± 0.52
	2	95.26 ^BX*^ ± 0.18	90.84 ^BW^ ± 0.32	−2.90 ^CW*^ ± 0.04	−0.68 ^CX**^ ± 0.02	18.76 ^CW**^ ± 0.04	19.04 ^CX*^ ± 0.13	19.56 ^W^ ± 0.48	21.14 ^bX^ ± 0.35	28.13 ^Bw^ ± 0.40	29.93 ^bx^ ± 0.29

WI—white index; YI—yellowing index; ^a^, ^b^, ^A^, ^B^, ^C^—differences between the percentage of WPC within a bacterial culture; ^a^, ^b^—significant differences at *p* ≤ 0.05; ^A^, ^B^, ^C^—significant differences at *p* ≤ 0.01; ^w^, ^x^, ^W^, ^X^—differences between days of storage; ^w^, ^x^—significant differences at *p* ≤ 0.05; ^W^, ^X^—significant differences at *p* ≤ 0.01; *, **—differences between bacterial cultures used within a percentage of WPC; *—significant differences at *p* ≤ 0.05; **—significant differences at *p* ≤ 0.01.

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
