# Peer review of "Changes in the Physicochemical Parameters of Yoghurts with Added Whey Protein in Relation to the Starter Bacteria Strains and Storage Time"

_animals, 2020, doi:10.3390/ani10081350_

Round 1

Reviewer 1 Report

Detailed notes and comments are included in the attached file.

Author Response

Dear Reviewer 1,

We appreciate the time and effort that you have dedicated to providing your valuable feedback on our manuscript. We are grateful for your insightful comments on the manuscript. We have been able to incorporate changes to reflect the suggestions provided in the review. We have used the „Track Changes” function in Microsoft Word. Here is a point-by-point response to the Reviewers’ comments and concerns:

L48 – „salivarius ssp.” was removed

L49 – „cm3” was replaced by „g”

L69-78– “The Introduction chapter lacks information on the health-promoting properties of whey proteins”.

Authors completed the information in this field (L64-73). Indeed, this part of text is now more precise. Whey proteins (consisting of individual proteins, i.e., α-lactalbumin, β-lactoglobulin, serum albumin, immunoglobulins, lactoferrin, lactoperoxidase and lysozyme) exert a well-documented positive effect on human health, including supporting the functioning of the cardiovascular, digestive and nervous systems, and demonstrating anticancer, antioxidative, antimicrobial (antibacterial and antiviral) and hypocholesterolemic effects. These proteins, as a uniquely rich and balanced source of amino acids (i.a., cysteine and tryptophan), possess a significant meaning for human nutrition. Occurrence of whey proteins exhibiting antimicrobial properties, i.e. immunoglobulins, lactoferrin, lactoperoxidase and lysozyme, in the diet is one of the factors determining the correct immune response. Moreover, these proteins, once partially digested, serve as a source of bioactive peptides with further physiological activities.

L99-110 – “It would be interesting to know how much of the whey is processed into WPC.” The main industrial processing of whey is drying (i.e. whey powder production), which is 70% of the annual production of whey. Small tonnage of WPC and to a lesser extent WPI (whey protein isolates) are produced every year, and the remaining permeate can be used to produce WPH (whey protein hydrolysates), protein fractions, lactic acid, bioethanol or lactose. It is estimated that about 20% of the whey is converted into WPC.

L116 – “Bulk milk from how many cows?”

Bulk milk was obtained from one farm where 23 Simmental cows were kept.

L116-117 – The indicated fragment of the text was deleted.

L121-122 – “The SH unit should be defined or converted to % lactic acid”

Potential acidity was determined in Soxhlet-Henkl’s degree (°SH) and expressed in the revised manuscript as lactic acid content, taking into account that 1 °SH = 0.0225% of lactic acid (Table 2).

L133-134 – “What was the recommendation of the manufacturer of the starter culture regarding its addition to milk. Was a freeze-dried or frozen culture used?”

We used the freeze-dried DVS bacterial cultures intended for yoghurt production from Chr. Hansen. Generally, the manufacturer recommends using 500U/2500 l of milk. For small packages it is 10U/50 l of milk. It is the smallest package possible to buy. This amount (0.15 g/l) was calculated and adjusted to own needs. We have also verified it with other scientific papers based on the vaccines from this manufacturer.

L131-133 – “In what way (mixed, homogenized?) and when (before pasteurization?) was WPC added to the milk? Was the mixture stable during such long pasteurization?”

WPC was added to the milk after heat treatment and cooling down to 40°C. The mixture was homogenized, and it was stable during the whole incubation and storage.

L138 – “What kind of plastic? PP or PS?”

We used the polypropylene (PP) plastic 100 ml containers.

L138-143 – “What was the incubation time? Did it take the same amount of time to reach pH 4.6 for all samples?”

Incubation should be stopped as soon as possible. For this purpose, we used ice water as well as cold air – research was carried out in an air-conditioned room. It is difficult to estimate the time clearly, but it did not last longer than 10 minutes. After cooling down, the samples were directly put to the storage. Samples with WPC reached pH 4.6 a little faster, i.e., incubation time was shorter by approx. 20-30 min.

L153 – “There is no description of the WHC determination methodology”

Methodology of WHC determination has been added to the Materials and Methods section (L163-168). Authors apologize for the oversight.

L155 – “(in °SH)” was removed.

L161– “References?”

Determinations were conducted basing on the standard PN-A-86061:2006. Milk and dairy products – Fermented milk. The references was added to the list.

L170-180 – “Texture parameters must be defined, how were they calculated?”;

“Gumminess and chewiness are mutually exclusive, rather gumminess should be reported as yoghurt is a semisolid food”;

“How? As it was set type of yoghurt it must be mixed before placed in the baker. The method of mixing should be defined as it affects the measured texture parameters.”

The methodology for determining texture parameters was thoroughly verified. Due to the large volume of the manuscript, the authors found it problematic to include an additional six tables. Only the most important parameters were presented, i.e., firmness (hardness), consistency (adhesiveness) and cohesive strength. The methodology was completed with the definitions of these parameters. Yoghurt samples were not transferred because they were prepared in a dedicated beaker.

L274-277 – “Results of this analysis are missed in the text.”

A detailed assessment of the flavour and aroma was added to the text of manuscript (L723-733).

L274-277 – “Which results were analysed one-way and which multi-way?”

One-way analysis of variance was applied for the following parameters: acidity, proximate chemical composition and hygienic quality of bulk milk used for yoghurt production, and multi-way for: active acidity, lactic acid content, proximate chemical composition, water activity, water-holding capacity, texture and colour parameters, and sensory evaluation of the yoghurts.

L317 – “Since whey proteins have buffering properties, their addition probably influenced the fermentation time (up to pH 4.6). So, were there any differences in fermentation times?”

Samples with WPC reached pH 4.6 a little faster, i.e., incubation time was shorter by approx. 20-30 min.

L323-324 – “It does not match the data in the table 3.”

The authors provided more precise values, i.e., decrease by an average 0.18 and 0.21, respectively for the strain.

L348-349 – “This is a false interpretation. The addition of whey proteins increases the buffering properties of milk, which influences the potential (titratable) acidity. The addition of lactose does not matter because the degree of its fermentation in fermented milks usually does not exceed 30%.”

The authors are deeply grateful for this valuable comment. This has been included in the text of manuscript.

L382-384 – “In the interpretation of the obtained results, it would be helpful to discuss the influence of the interactions of the examined factors”

The manuscript was completed with the indicated results. The interactions of the examined factors, i.e., WPC content and day of storage, proved to be significant at p ≤ 0.05. However, the inclusion of type of bacterial cultures additionally in the analysis indicated no significant interactions of these factors on the lactic acid content.

L401-402 – “To support this thesis, it would be worth including data on the chemical composition of the WPC (manufacturer's data?) used in the experiment.”

The manuscript was completed with the indicated results.

L405-406 – “How do the authors explain this phenomenon? Were the yoghurt packages sealed?”

The yoghurt packages were sealed in order to limit the evaporation and pollution from the environment. The yoghurts were opened on the day of analysis. Perhaps it resulted from the activity/specifics of bacteria used.

L408/Table 5– “How do the authors explain such large differences in protein content between yoghurts without WPC (4.16-4.26%) and the milk from which they were made (3.74)?”

The differences in protein content between milk and yoghurts without WPC were assuredly due to the activity of bacteria used.

L581-617 – “In the opinion of the reviewer, these results should be presented more precisely or removed from the work.”

The results have been presented more precisely. Additional table – Table 10 were placed in the text.

L641-642 – “?”

These values were for a solid non-fat (SNF) content (9.07%) and fat (1.50%), however, the authors removed them from the presented version of the manuscript.

L723-733 – “The results of the sensory evaluation should be discussed in more detail.”

The authors discussed in more detail the sensory evaluation of yoghurts.

L717, 721 – “In this form, the figure is unclear. The descriptions on the axes are illegible”

The authors changed the form of the Figures 4 and 5.

L755-756 – “In the opinion of the reviewer, this conclusion is too optimistic and not entirely consistent with the obtained results, which indicate a significant reduction of sensory quality of yoghurt with WPC (mainly beetwen 14-28 days), especially the two most important features, i.e. flavour and aroma.”

The indicated fragment of the text has been changed.

L756 – “A whey product like WPC 80% is not a "problematic waste", but valuable product.”

The authors are deeply grateful for this valuable comment.

L766 – “References should be prepared according to the journal's requirements.”

The authors adjusted references to the journal’s requirements.

Your sincerely,

Authors

Reviewer 2 Report

Obtaining yogurts with improved functional and sensory characteristics a priority both due to the diversification of assortments but especially to cover the largest segments of consumers. 

Dairy products and especially yogurt are staple foods for all age groups and improving their qualities also determines a beneficial action for consumers .

I believe that research brings important benefits i the field of quality food production. 

Author Response

Dear Reviewer 2,

We appreciate the time and effort that you have dedicated to providing your valuable feedback on our manuscript.

Your sincerely,

Authors

Reviewer 3 Report

Review

The manuscript is written in a clear way. Also the topic is interesting and helpfull for dairy industry. Additions of bacterial strains to dairy products are known and often described. The work is new, especially in detailed analysis of the effects of the addition AWP.

I have one question to authors:

Small differences were often significant in manuscript. Was repeated measurement used during collecting data and also statistical analysis?

20-21 A different sentence should be chosen - the experiment does not provided results on the positive impact of WPC outside the dairy industry.

31 - not all described effects of the additive can be described as positive

FIG. 4 and 5 ….. illegible labels

Tab. 4 and 10...some raws in table are shifted, could be confusing

The article brings new findings, so I recommend to publish the manuscript with a minor revision.

Author Response

Dear Reviewer 3,

We appreciate the time and effort that you have dedicated to providing your valuable feedback on our manuscript. We are grateful for your insightful comments on the manuscript. We have been able to incorporate changes to reflect the suggestions provided in the review. We have used the „Track Changes” function in Microsoft Word. Here is a point-by-point response to the Reviewer’ comments and concerns:

„Small differences were often significant in manuscript. Was repeated measurement used during collecting data and also statistical analysis?”

All measurement taken during collecting data were repeated and also statistical analysis. A total of 60 yoghurts were analysed: 20 without WPC (control) and 40 with the addition of WPC.

L20-21 – „A different sentence should be chosen - the experiment does not provided results on the positive impact of WPC outside the dairy industry.”

The authors are deeply grateful for this valuable comment. This has been included in the text of manuscript.

L31 – „Not all described effects of the additive can be described as positive”

The word „positive” has been changed to „significant”.

L717, 721 – “FIG. 4 and 5 ….. illegible labels.”

The authors posted the Figures 4 and 5 in a new form.

„Tab. 4 and 10...some raws in table are shifted, could be confusing”

The authors tried to change the distribution in the tables. Although, it will probably be corrected at the stage of printing the text of manuscript, i.e. on the editorial stage.

Your sincerely,

Authors